# In Vivo Assessments of Mesoblastic Nephroma (Ne/De) and Myelomonoblastic Leukaemia (My1/De) Tumour Development in Hypercholesterolemia Rat Models

**DOI:** 10.3390/ijms232113060

**Published:** 2022-10-27

**Authors:** Zita Képes, Alexandra Barkóczi, Judit P. Szabó, Ibolya Kálmán-Szabó, Viktória Arató, Ildikó Garai, Péter Árkosy, István Jószai, Ádám Deák, István Kertész, István Hajdu, György Trencsényi

**Affiliations:** 1Division of Nuclear Medicine and Translational Imaging, Department of Medical Imaging, Faculty of Medicine, University of Debrecen, Nagyerdei St. 98, H-4032 Debrecen, Hungary; 2Department of Urology, Faculty of Medicine, University of Debrecen, Nagyerdei St. 98, H-4032 Debrecen, Hungary; 3Doctoral School of Clinical Medicine, Faculty of Medicine, University of Debrecen, Nagyerdei St. 98, H-4032 Debrecen, Hungary; 4Gyula Petrányi Doctoral School of Clinical Immunology and Allergology, Faculty of Medicine, University of Debrecen, Nagyerdei St. 98, H-4032 Debrecen, Hungary; 5Department of Oncology, Faculty of Medicine, University of Debrecen, Nagyerdei St. 98, H-4032 Debrecen, Hungary; 6Department of Operative Techniques and Surgical Research, Faculty of Medicine, University of Debrecen, Nagyerdei St. 98, H-4032 Debrecen, Hungary

**Keywords:** butter and cholesterol-rich (BCR) diet, [^18^F]F-FDG PET/MRI, hypercholesterolemia, lipids, myelomonoblastic leukaemia (My1/De), mesoblastic nephroma (Ne/De), standardised uptake value (SUV)

## Abstract

Given the rising prevalence of lipid metabolic disorders and malignant diseases, we aimed to establish an in vivo hypercholesterinaemic tumour-bearing rat model for the induction and assessment of these conditions. A normal standard CRLT/N, 2 (baseline),- or 4 (2 + 2, pretreated)-week-long butter and cholesterol rich (BCR) diet was applied to mesoblastic nephroma (Ne/De) and myelomonoblastic leukaemia (My1/De) tumour-bearing and healthy control Long—Evans and Fischer 344 rats. The beginning of chow administration started in parallel with tumour induction and the 2 weeks of pre-transplantation in the baseline and pretreated groups, respectively. Fourteen days post-inoculation, the measurement of lipid parameters and [^18^F]F-FDG PET/MRI examinations was executed. The comparable lipid status of baseline healthy and tumorous rats proves that regardless of tumour presence, BCR-based hypercholesterolemia was achieved. A higher tumour mass among pretreated tumorous animals was found when compared to the control groups (*p* < 0.05, *p* < 0.01). Further, a visually greater [^18^F]F-FDG accumulation was observed in pretreated BCR tumorous animals; however, the quantitative data (SUV_mean_: 9.86 ± 0.98, 9.68 ± 1.24; SUV_max_: 19.63 ± 1.20; 17.56 ± 3.21 for Ne/De and My1/De, respectively) were not statistically significantly different from those of the CRLT/N tumorous rats (SUV_mean_: 8.40 ± 1.42, 7.22 ± 1.06 and SUV_max_: 15.99 ± 2.22, 12.46 ± 1.96 for control Ne/De and My1/De, respectively). Our model seems to be appropriate for simultaneously investigating hypercholesterolemia and cancer in the same rat.

## 1. Introduction

Since lipid metabolic disturbances and malignant diseases represent one of the largest yet-known non-infectious pandemics worldwide, efficient treatment regimens and preventive measures must be established in order to address these pertinent global threats. Experimental nuclear medical in vivo animal models, which ensure a valuable scenario for the thorough understanding of the development of these diseases, may serve as potential tools for long-term radiopharmaceutical development and drug testing.

Dyslipidaemias stemming from various genetic and environmental factors could ultimately result in pathologically elevated plasma lipid levels [1,2,3]. Hypercholesterolemia accounts for one of the most staggeringly threatening manifestations of dyslipidaemia [1]. Given the increasing prevalence of lipid alterations and neoplastic diseases, an immense number of research studies has been spawned that already support the connection between these abnormalities.

The literature data support the fact that serum cholesterol level-associated chronic inflammation plays a crucial role in the appearance of malignancies [4,5]. In a recent preclinical study, a prolonged cholesterol-rich diet was reported to trigger the spontaneous development of hepatocellular carcinoma related to non-alcoholic fatty liver disease (NAFLD-HCC) in mice [6]. Further, the literature data support the notion that diet and low-density lipoprotein receptor (LDL-R) deficiency-induced hypercholesterolemia promoted urinary bladder cancer stemness and enlargement [7].

Preclinical, clinical, and epidemiological studies support the association of the harmful lifestyle elements in connection with lipid abnormalities and neoplasms. Malnutrition—in terms of both quality and quantity—high saturated fat intake, lack of physical activity, and excess alcohol consumption are all acknowledged risk factors of these diseases [8,9,10,11]. Considering the overwhelming financial, social, and economic burden that these health concerns impose on societies, the need for the introduction of investigational models that excel in the simultaneous assessment of both tumorous diseases and lipid abnormalities is highlighted. Ongoing preclinical research studies have already evaluated the efficacy of different drug-to-be molecules, such as statins, that could be suitable for disease prevention and management; however, some shortcomings of these experiments do need to be re-evaluated [12]. The length of the studies, the large number of involved and sacrificed animals, as well as the rising costs are amongst the most pressing points to consider. Consequently, the implementation of such a preclinical model system, which enables the rapid as well as cost-effective testing of novel drug candidates that will be applied to only a small number of animals, is one of the priorities of modern science. In 2009, Trencsényi et al. managed to create a rat animal model system that provided adequate circumstances for considerable tumour growth and metastasis formation within two weeks [13]. Prior literature reports are also available regarding the investigation of drug effects on the tumour development of hypercholesterinaemic rats [14,15].

Therefore, in the present study, we aimed to propose a translational in vivo animal model that enables the parallel induction and assessment of malignancies and hypercholesterolemia. Additionally, we also intended to make this model suitable for the evaluation of effective anti-dyslipidaemic and anti-tumour molecules before human application.

## 2. Results

### 2.1. Effect of Butter and Cholesterol-Rich (BCR) Diet on Healthy Rats

In the case of healthy rats, the results of the clinical chemistry blood tests of the animals kept on a standard Charles Rivers Laboratories feed/normal (CRLT/N, group 1) diet and on a BCR diet for either two (group 2) or four (group 3) weeks were compared. Based on the results of the blood tests, we can conclude that both the total cholesterol and low-density lipoprotein cholesterol (LDL-cholesterol) levels of the rats on a baseline BCR diet were significantly elevated compared to those fed with the standard chow (*p* < 0.01; Figure 1A). Therefore, it can be summarised that in applying BCR animal feed, we managed to achieve hypercholesterolemia (Figure 1A). Further, the high-density lipoprotein cholesterol (HDL-cholesterol) levels of the two animal groups did not differ significantly (*p* < 0.05, Figure 1A).

By analysing the blood test results of group 3 (i.e., the BCR-pretreated group) where the BCR diet was administrated for 4 weeks, we revealed significantly increased cholesterol and LDL-cholesterol levels when compared to the CRLT/N group of healthy rats (*p* < 0.05, Figure 1A). However, relatively higher cholesterol and LDL-cholesterol levels were observed in the BCR-pretreated group in comparison with the baseline BCR diet group, but this difference was not statistically significant (*p* < 0.05). The HDL levels did not show meaningful differences in the BCR-pretreated cohort when we compared it with the two other groups (Figure 1A). Further, the two different diets led to changes in the macroscopic appearance of the liver parenchyma (Figure 1B,C). Figure 1B demonstrates a healthy liver, whereas in Figure 1C, a representative autopsy picture of a steatotic liver, induced by BCR animal feed, can be identified.

### 2.2. Effect of BCR Animal Diet on the Blood Cholesterol Levels of Mesoblastic Nephroma (Ne/De)-Transplanted F-344 Rats

The effects of the BCR animal feed were also assessed in case of Ne/De transplanted tumour-bearing F-344 rats. Three groups of Ne/De tumorous animals were compared in this part of the experiment: Ne/De tumorous rats on a standard (CRLT/N) diet for two weeks (group 4), Ne/De tumorous rats administered with a baseline BCR feed for two weeks (group 5), and Ne/De tumour-bearing BCR-pretreated rats administered with a BCR diet for four weeks (group 6). In the case of group 5 (baseline BCR diet), the beginning of the administration of the diet coincided with the time of the transplantation, whereas in group 6 (BCR-pretreated group), the rats were administered a BCR diet 2 weeks prior to the tumour induction; further, this diet was maintained for two more weeks post-incubation.

Healthy rats on a standard diet in group 1 presented physiological blood test values (shown in Figure 1A) and comparable results were found in group 4 for Ne/De tumorous rats on a standard diet (Figure 2). As for healthy rats in group 2 (baseline BCR diet), we measured the lipid parameters corresponding to the effects of the BCR feed (Figure 1A). The blood test results of the tumorous rats of group 4 (standard CRLT/N diet, Figure 2) were largely similar to those of healthy rats fed a BCR diet for 2 weeks (Figure 1A). Consequently, it can be concluded that the presence of a tumour does not influence the development of BCR-related hypercholesterolemia (Figure 2).

Significantly increased cholesterol and LDL-cholesterol levels were defined in the pretreated Ne/De tumour-bearing group of rats (group 6) compared to both the tumorous rats on the standard diet (group 4) and those on the baseline BCR (group 5) rodent chow (*p* < 0.05, Figure 2). No statistically remarkable differences were found regarding the HDL values of either group (Figure 2).

### 2.3. Growth of Ne/De Tumours in Normolipidaemic and Dyslipidaemic Rats

We further aimed to investigate whether a BCR diet affects the development of the tumours. During the autopsy of the tumorous animals, both the right and the tumour-bearing left kidneys of the rats were removed. Following weight measurement, the tumour mass was determined by subtracting the mass of the right kidney from that of the enlarged left kidney. Comparing the results of the CRLT/N (group 4) and baseline BCR diet (group 5) groups, we can conclude that a BCR diet influences tumour growth, as higher tumour mass values were obtained; however, this difference was not statistically significant (*p* < 0.05; Table 1). In contrast, in group 6—where the rats received a BCR diet for 2 weeks prior to the tumour induction, and for an additional 2 weeks post-transplantation—we demonstrated that the tumorous left kidneys of the BCR-pretreated animals had notably higher weights compared to those of the tumorous CRLT/N group (*p* < 0.05 and *p* < 0.01; seen in Table 1).

### 2.4. 2-Deoxy-2-[^18^F]Fluoro-D-Glucose Positron Emission Tomography/Magnetic Resonance Imaging ([^18^F]F-FDG PET/MRI) Examinations of Ne/De Tumours in Normolipidaemic and Dyslipidaemic Rats

To confirm the presence of tumours and to monitor their growth, [^18^F]F-FDG PET/MRI examinations were performed in the groups of the Ne/De tumour-bearing rats that were kept on a standard (group 4) and baseline BCR (group 5) diet, as well as in the BCR-pretreated group of rats (group 6). A quantitative analysis of [^18^F]F-FDG accumulation, characterising glucose metabolism, was also conducted. Although slightly increased tumour standardised uptake values (SUV) were observed in group 5 (SUV_mean_: 8.69 ± 0.74 and SUV_max_: 16.93 ± 1.50) and group 6 (SUV_mean_: 9.86 ± 0.98 and SUV_max_: 19.63 ± 1.20) when compared to group 4 (SUV_mean_: 8.40 ± 1.42 and SUV_max_: 15.99 ± 2.22), no statistically noticeable difference could be identified between the three different groups of the Ne/De tumorous animals regarding their [^18^F]F-FDG uptake (Figure 3). In addition, upon visual assessment an enhanced tumour tracer uptake in the BCR-pretreated group (group 6) was found; however, this was not statistically significant either (*p* < 0.05). These results support the notion that BCR animal feed does not exert a meaningful effect on either tumour metabolism or its glucose metabolism.

### 2.5. Impact of BCR Diet on Blood Cholesterol Levels of Myelomonoblastic Leukaemia (My1/De) Tumour-Bearing Long–Evans Rats

The effect of the BCR diet on blood cholesterol levels was also assessed in the My1/De tumour-bearing groups. Similar to the case of the My1/De group 5 (baseline BCR diet)—in regard to the experiments with the Ne/De tumorous animals—the beginning of the administration of the diet coincided with the time of the My1/De tumour transplantation, whereas in the My1/De group 6 (BCR-pretreated group), the start of nutrition began 2 weeks prior to the My1/De tumour induction; further, the diet was maintained for an additional two more weeks. In contrast with the standard diet, we managed to induce dyslipidaemia (Figure 4) in both BCR-fed My1/De animal groups. The blood test results of the rats of group 1 (healthy rats, standard diet) appeared to be normal (Figure 1A). Further, almost identical values were obtained in the case of the animals in group 4 (tumour-bearing rats, standard diet, Figure 4). As formerly mentioned, regarding the laboratory parameters of the healthy rats in group 2 (baseline BCR diet, Figure 1A), the lipid values corresponded to the effects of the BCR diet. The lipid profile of the My1/De tumorous rats of group 5 (baseline BCR diet) was greatly similar to that of the healthy rats kept on BCR feed for 2 weeks. Therefore, our results prove that the presence of tumours does not influence the BCR-diet-based induction of hypercholesterolemia (Figure 4).

The pretreated animals of group 6 demonstrated noticeably elevated cholesterol and LDL-cholesterol levels compared to the other two My1/De tumour-bearing animal groups (*p* < 0.05). However, no statistically significant differences were found regarding the HDL values of either group (*p* < 0.05; Figure 4).

### 2.6. Growth of My1/De Tumours in Normolipidaemic and Dyslipidaemic Rats

For the purposes of assessment on the influence of the BCR diet on tumour growth, tumour mass measurement was completed in My1/De tumour-bearing Long–Evans rats as well. After the comparison of the tumour weights of the My1/De tumorous CRLT/N (group 4) and baseline BCR diet (group 5) groups, we found relatively higher values in the group fed with the baseline BCR diet; however, this difference was not statistically significant (*p* < 0.05; seen in Table 2). In contrast, noticeably higher My1/De tumour weights were observed in group 6 (i.e., the pretreated BCR group) than in the CRLT/N group (*p* < 0.05 and *p* < 0.01; demonstrated in Table 2).

### 2.7. [^18^F]F-FDG PET/MRI Examinations of My1/De Tumours in Normolipidaemic and Dyslipidaemic Rats

My1/De tumour-bearing rats fed with standard (group 4), baseline BCR (group 5), and pretreated BCR (group 6) diets underwent [^18^F]F-FDG PET/MRI examinations in order to verify the presence of the subrenally growing tumours. Based on the qualitative analysis of the decay-corrected PET/MRI images, increased tumour volume and [^18^F]F-FDG uptake were observed in the group fed with a baseline BCR diet (group 5, Figure 5B) compared to the group on the standard diet (group 4, Figure 5A). Although more intense [^18^F]F-FDG accumulation was visually experienced in the BCR-pretreated group of animals (group 6, Figure 5C), statistically speaking, it was not meaningful. Quantitative image analysis revealed the lowest [^18^F]F-FDG accumulation in group 4 (SUV_mean_: 7.22 ± 1.06 and SUV_max_: 12.46 ± 1.96), followed by group 5 with SUV_mean_ and SUV_max_ values of 7.37 ± 1.15 and 13.93 ± 3.10, respectively (Figure 5D). The highest values were observed in the BCR-pretreated group (SUV_mean_: 9.68 ± 1.24 and SUV_max_: 17.56 ± 3.21); however, similar to the experiments with the Ne/De tumours, we did not detect any significant differences either in regard to the [^18^F]F-FDG accumulation of the malignancies of the different My1/De tumour-bearing rats (*p* ≤ 0.05; Figure 5D).

### 2.8. [^18^F]F-FDG PET/MRI Investigation on Metastases in Tumour-Bearing Dyslipidaemic Rats

In the last part of our experiments, the formation of metastases was investigated in both the baseline and pretreated groups (groups 5 and 6) of BCR-fed Ne/De and My1/De tumour-bearing rats. Qualitative analysis of the decay-corrected PET/MRI images revealed a more frequent development of liver metastases with enhanced [^18^F]F-FDG accumulation in the BCR-pretreated group of tumour-bearing animals (group 6; Figure 6B right and Figure 6D right). In addition, the thoracic parathymic lymph nodes were also depicted with increased radiopharmaceutical uptake (shown in Figure 6A right and Figure 6C right). In contrast, in the baseline BCR-fed groups of Ne/De and My1/De tumour-bearing rats (group 5), a lower number of moderately accumulating metastases were detected (demonstrated in Figure 6).

## 3. Discussion

Dyslipidaemia and malignant diseases represent major health concerns in today’s world [1,16]. Sobering data, illustrating the rising prevalence of these two diseases, warrant the critical need for the introduction of novel drug candidates and diagnostic methods that may be able to revolutionise the treatment and diagnostics of these chronic disorders [1,16]. As animal models are meant to be appropriate for the authentication of the safety and the efficacy of new anti-cancer and anti-dyslipidaemic drugs-to-be, we were authorised to develop an experimental animal model that enables drug evaluation in tumour-bearing hypercholesterinaemic rats.

The preclinical literature data highlight the success of animal models regarding analyses of the effects of medications in one single disorder, such as in neoplastic diseases or lipid abnormalities. Kertai and colleagues intended to appraise fluvastatin-associated tumour growth inhibition, as well as metastasis development, in a rat animal model [17]. Six weeks after the prior administration of fluvastatin to FLF1 rats, He/De (epithelial hepatocellular) tumour cells were transplanted under the left renal capsule of the animals. Provided that the lymph node metastases were assumed to be more sensitive to fluvastatin-induced effects, when compared to the primer malignancies, this experimental model did seem to be promising in regard to treatment evaluation. In another study, hypercholesterinaemic female Fischer rats were disseminated in order to illustrate fluvastatin-related myopathy [18]. Although the applied therapy led to a decrease in the cholesterol levels of the animals, rising creatine kinase (CK) parameters showed statin-based muscle side effects. These findings also draw attention to the appropriateness of preclinical investigational systems for the profound evaluation of the mechanisms of action of different drugs.

With the aim of the parallel assessment of the dual—anti-tumour and lipid lowering—effects of fluvastatin (Lescol^®^, Novartis, Switzerland, Basel), Kertai and his research team established an animal model using He/De implanted male and female F1 hybrids of Long–Evans and Fischer344 rats that were kept on a cholesterol-rich diet (CRD) [15]. Prior research studies dealing with the anti-tumour and preventive effects of fluvastatin revealed that, depending upon the applied therapeutic regime, fluvastatin exerts both chemopreventive and therapeutic effects on the development of subrenally growing hepatocarcinoma cells (He/De) when it is implemented in FLF1 hybrid rats [19]. Further, Kertai and colleagues strengthened the suitability of the FLF1 experimental rat model in the simultaneous investigation of the efficacy of anti-cancer and anti-hypercholesterolemia medications in the same rat [14]. Briefly, in their series of experiments, FLF1 rats were fed with a CRD diet for 6 weeks. Then, subrenal transplantation of He/De tumour cells and fluvastatin treatment administration started on the 21st day of the study. At the endpoint of the assay, the weight of the tumour mass and serum lipid values were measured.

Taking into account the strong association between and frequent coexistence of malignant diseases and lipid disorders, we were keen to establish a preclinical rat model that enables the parallel exploration of drug effects in tumour-bearing hypercholesterinaemic rats.

Two types of chemically induced tumorous models were used in our study. The term “mesoblastic nephroma” (Ne/De) refers to the pathological proliferation of the connective tissue cells of the kidneys, both in rats and in humans. The tumour is characterised by extremely rapid growing potential with extensive metastatisation throughout the body. An intraperitoneal injection of a physiological saline solution of 125 µg of N-nitrosodimethylamine into newborn rats triggers the formation of Ne/De tumours. After a lag period of 5–7 months, in 90% of the cases, the tumour can be identified. My1/De is one of the subtypes of the leukaemias that is featured with a pathological division of the stem cells—the myelomonoblasts—of the bone marrow. This is a tremendously malignant and incurable disease, despite the life-prolonging accessible treatment, which entails a lot of suffering. By applying the method of Huggins and Sugiyama, i.v. injection of a solution of 3 × 40 mg/body weight of 7–12 dimethylbenz(a)anthracene was used to induce My1/De tumour formation in rats. In this case, the duration of the latent period was, similarly, 5–7 months.

Dyslipidaemia in rats was first induced by Fillios and colleagues, whereby they applied a cholesterol–cholic acid and thiouracil-containing diet that was supplemented with butter provided by Thomas and Hartroft [20]. Although this modified animal feed proved to be effective, the two-month-long hypercholesterolemia induction posed a challenge for the researchers. Later, Padra and colleagues reported that a diet solely rich in cholesterol causes only slightly elevated plasma cholesterol levels, but adding butter to it—which has high cholesterol as well as saturated fatty acid content—leads to significantly increased cholesterol values [21]. Given these findings, the development of hypercholesterolemia was achieved over only two weeks in our experimental animals with the application of a BCR diet. In addition, to engrave cholesterol absorption—such that the cholesterol level lowering effects of the thyroid hormones can be inhibited—and to satisfy the amino acid need of the rats and to elevate blood pressure, sodium-cholate, methyl-thiouracil, casein, and Sós’ SM8 salt mixtures were, respectively, added to our formulations of the BCR diet. Padra et al. found that the animals had more than 20, 30, and 6 times higher cholesterol, LDL/HDL ratios, and triglyceride levels, respectively, following two weeks of BCR diet administration [21]. In line with their findings, we also recorded that due to the effects of the BCR animal feed, the serum total cholesterol levels of the healthy F-344 rats were elevated from 1.29 ± 0.31 mmol/L to 6.51 ± 1.01 mmol/L, whereas the LDL-cholesterol values showed an increase from 0.22 ± 0.08 mmol/L to 4.05 ± 0.67 mmol/L (Figure 1). Based on the publication of Padra and colleagues, because of the rise in total cholesterol—particularly in LDL-cholesterol levels—as well as the change in lipid ratios, using the term “dyslipidaemia” is more adequate than “hyperlipidaemia” [21]. In the present study, dyslipidaemia, characterised by changes in cholesterol values similar to those of Padra’s study, was successfully induced by a BCR diet in all groups of healthy and tumour-bearing rats. Regarding the lipid profile of the healthy rats, no statistically significant difference could be found between the three different examined groups, i.e., the standard animal diet (group 1), baseline BCR diet (group 2), and BCR-pretreated (group 3) animals. In contrast, tumorous Ne/De and My1/De rats receiving BCR pre-treatment (group 6) showed the most elevated cholesterol and LDL-cholesterol values compared to the corresponding tumour-bearing animals kept on either baseline BCR (group 5) or normal diet (group 4, Figure 2 and Figure 4). We may draw the conclusion that although we managed to induce BCR-based hypercholesterolemia regardless of the duration of nutrition, lengthened administration times resulted in more pronounced lipid level changes in case of the tumorous animals. The heterogeneity of the examined animals, and their individual genetic characteristics, may be the reason behind why we observed significantly elevated lipid levels only in the tumorous groups of the BCR-pretreated animals when compared to the other two cohorts of the tumour-bearing rats; further, no statistically remarkable distinction could be observed within the healthy rat groups. In addition, this may be of clinical importance in the evaluation of long-term drug effects. Relatively short treatment regimens may explain why no considerable changes occurred regarding HDL levels.

According to our results, no statistically significant variance was found between the tumour mass of the CRLT/N tumorous (group 4) and tumour-bearing animal groups kept on the baseline BCR diet (group 5). However, a considerably higher tumour mass was found in the BCR-pretreated tumorous groups (group 6) compared to the tumour weight of rats on the standard diet, which reveals that the BCR diet has a time and regimen-dependent effect on tumour development (Table 1 and Table 2). This finding supports the notion that tumour growth is markedly associated with the cholesterol content of the consumed diet, thereby evidencing the potential efficacy of lipid-lowering in the prevention of tumour formation and progression.

When comparing the growth of the two tumours, the Ne/De tumours appeared to be larger than the My1/De neoplasms following two weeks post-SRCA transplantation, regardless of the applied diet. This outcome is strengthened by both the MRI images and the results of the tumour mass measurements. The quantitative evaluation of in vivo [^18^F]F-FDG PET/MRI-based tumour glucose metabolism also showed higher SUV values in the Ne/De tumours than were found in the My1/De neoplasms. This refers to the enhanced glucose uptake required for the more intense tumour growth rate. Although meaningful differences were not observed regarding the quantitative analysis of the [^18^F]F-FDG tracer accumulation (both SUV_mean_ and SUV_max_ values) of the three different groups of Ne/De and My1/De tumorous rats, greater radiopharmaceutical uptake upon visual assessment in the case of BCR-pretreated tumour-bearing animals (group 6) may suggest some associations between tumour growth and lipid status (Figure 3 and Figure 5). In the future, unbiased studies with longer pretreatment times are warranted in order to fully strengthen this hypothesis.

Finally, in the BCR-pretreated groups of rats (group 6)—which are characterised by the highest levels of lipid parameters—more frequent metastases formation with enhanced metabolic activity was triggered. As such, we hypothesise that tumour metastatisation and the present metabolic activity of the metastases are strongly dependent upon the actual lipid status of the rats (Figure 6). Further, we assume that prolonged BCR preincubation times may result in the appearance of secondary tumours in other localisations besides the liver and the parathymic lymph nodes.

Although the literature data support the notion that cholesterol metabolites—such as cholesteryl esters, 5,6-epoxycholesterols (as direct or indirect tumour promoters), and 27-hydroxycholesterol (involved in the growth of oestrogen receptor (ER)-positive breast cancer cells)—are associated with tumorigenesis, the assessment of the detailed cholesterol metabolites on tumour growth was not among the aims of the present study [22,23,24,25,26,27,28]. However, preclinical hypercholesterinaemic tumour model studies may also bring us closer to an in-depth understanding of the role of these metabolites in carcinogenesis.

Overall, in our study, dyslipidaemia was successfully and safely induced in rats. Neither of the examined tumours were seen to influence the lipid-raising effect of the BCR diet. Further, we assert that the diet did not affect tumour growth either. Consequently, the currently applied animal model seems to be suitable to simultaneously test the impacts of promising anti-dyslipidaemic and anti-tumour molecules on the same rats. Moreover, this hypercholesterinaemic tumour model system holds rapid and cost-effective future potential for drug trials using a relatively small number of experimental animals.

## 4. Materials and Methods

### 4.1. Cell Culturing

Chemically induced Ne/De and My1/De cells were utilised for tumour formation in our research. Cell cultivation occurred at a 37 °C temperature under a 5% CO_2_ atmosphere in a Dulbecco’s modified Eagle medium (DMEM, Merck, Darmstadt, Germany) supplemented with 1% (*vol*/*vol*) antimycotic and antibiotic solution (Merck, Darmstadt, Germany), and 10% (*vol*/*vol*) heat-inactivated foetal bovine serum (FBS, Merck, Darmstadt, Germany). Due to their rapid growth, every second day, a passage was conducted in order to reach the appropriate cell count.

### 4.2. Experimental Animals

Thirty-four Long–Evans (200–220 g weighted) and 34 Fischer 344 (180–200 g weighted) female rats were used in our study, which were kept in conventional laboratory circumstances at a temperature of 26 ± 2 °C with 55 ± 10% humidity and artificial lighting with a circadian cycle of 12 h. The animals were fed and provided with tap water ad libitum. The experiments were performed in compliance with the criteria of the Ethics Committee for Animal Experimentation of the United Kingdom with the permission of the Ethics Committee for Animal Experimentation of the University of Debrecen (approval number: 21/2017/DEMÁB; 3-1/2014/DEMÁB). The detailed subclassifications of the experimental animals are demonstrated in Table 3.

The administration of the BCR diet in group 6 (BCR-pretreated group) started 2 weeks prior to tumour induction, and this diet was maintained for two more weeks post-implantation.

Groups 1, 2, 4, 5, and groups 3 and 6 consisted of six and five animals, respectively, both in the case of Long–Evans and Fischer 344 rats.

### 4.3. Animal Feed

The CRLT/N animal feed was produced by Charles River (Charles River Ltd., Gödöllő, Hungary) and was used for the purposes of standard animal nutrition (as shown in Table 4), while a modified BCR laboratory chow was used for the purposes of cholesterol-rich animal feeding (demonstrated in Table 5). The basis of the CRLT/N animal feed was such that it was supplemented with casein, Na-cholate, and Sós’ SM8 salt mixture in order to contribute to the amino acid needs of the animals to ease cholesterol absorption, and to also elevate the blood pressure of the animals, respectively [29]. Since butter contains cholesterol, as well as saturated fatty acids, and the administration of cholesterol alone only slightly affects the cholesterol levels of the animals, we also added butter to the mixture to ensure a sufficient increase in blood cholesterol levels. In the case of the animals of group 5, the administration of a high-cholesterol BCR diet started on the day of the transplantation, while we began to feed the animals of group 6 with BCR chow 2 weeks prior to tumour induction.

### 4.4. Surgical Procedure

We aimed to implant cells on GelasponR disks under the left renal capsule of both the Fischer 344 and Long–Evans rats [30]. Firstly, 1 mm-thick disks with a 4 mm diameter were created from Gelaspon^R^ sheets (Germed, Rudolstadt, Germany) and sterilised. During the experiments, 1 × 10^6^ Ne/De and 1 × 10^6^ My1/De cells placed on Gelaspon^R^ disks (Bausch & Lomb, Vaughan, ON, Canada) in a 10 μL physiological saline solution (0.9% NaCl solution) and were subrenally transplanted into the experimental animals. With the intraperitoneal (*i.p.*) administration of 60 mg/body weight of Nembutal (Pentobarbital sodium) the rats were anaesthetised. Then, we shaved the fur off their left lumbar region and this area was disinfected. Following the incision of the skin of the lumbar territory, the retroperitoneum was opened and the disks containing the tumour cells were implanted under the capsule of the exposed kidney. The muscle layer was then sutured, and the wounds were closed with surgical stitches. Autopsies and in vivo examinations were executed two weeks later.

### 4.5. Blood Tests

The lipid profile (cholesterol, HDL-cholesterol, and LDL-cholesterol) of the rats was assessed from the intravenous (*i.v.*) serum samples of the animals. Specimen taking occurred two weeks after the start of feeding in the groups receiving the normal standard and the baseline BCR diet, and four (2 + 2) weeks after the beginning of nutrition in the case of the BCR-pretreated animals. The methods of Roeschlau and Allain (1974), Sugiuchi and colleagues (1995), and Sugiuchi et al. (1998) were applied for the measurement of these parameters on Roche/Hitachi cobas C systems (Roche Diagnostics Limited, Burgess Hill, West Sussex, UK), respectively [31,32,33]. The lipid fractions were determined in mmol/L.

### 4.6. A case of [^18^F]F-FDG PET/MRI: In Vivo PET/MRI Examinations

Fourteen days after SRCA-based tumour induction, 8.0 ± 0.4 MBq [^18^F]F-FDG was *i.v.* administered to the experimental animals via the lateral tail vein. Fifty minutes postinjection, T1 MRI and whole-body PET images were conducted (a 10 min acquisition was applied in all bed positions) using a preclinical PET/MR system (nanoScan PET/MRI 1T, Mediso Ltd., Budapest, Hungary). During the PET/MRI examinations, Forane-induced inhalation anaesthesia was applied with a dedicated small animal anaesthesia device (Tec3 Isoflurane Vaporizer, Eickemeyer Veterinary Equipment, Luton, UK) using 3% isoflurane (Forane, AbbVie, Budapest, Hungary; OGYI-T-1414/01), 0.4 L/min O_2_, and 1.4 L/min N_2_O.

During the evaluation of the PET data, the subsequent three values were determined:

(a)SUV (g/mL), which determines the concentration of the tracer within the region of interest, that is, the volume of interest (VOI) based on the administered dose and the weight of the animal.


$$\mathrm{SUV}=\frac{\mathrm{concentration}\;\mathrm{within}\;\mathrm{the}\;\mathrm{VOI}\;\left[\frac{\mathrm{MBq}}{\mathrm{mL}}\right]}{\mathrm{administered}\;\mathrm{dose}\left[\mathrm{MBq}\right]/\mathrm{body}\;\mathrm{weight}\;\left[g\right]}$$


(b)SUV_mean_, which is the average activity concentration of the VOI.(c)SUV_max_, which refers to the voxel with the highest activity within the VOI.

### 4.7. Statistical Analyses

Significance was calculated via Student’s two-tailed t-test, two-way ANOVA, and the Mann–Whitney rank-sum tests. The significance level was set at *p* ≤ 0.05 unless otherwise indicated. A commercial software package, MedCalc 18.5 (MedCalc Software, Mariakerke, Belgium), was used for all statistical analyses. Data are presented as the mean ± SD of at least three independent experiments.

## 5. Conclusions

Baseline (2-week long) or pretreated (2 + 2-week long) BCR diet-associated dyslipidaemia was induced in F-344 and Long–Evans rats. The total cholesterol and LDL-cholesterol levels of the rats fed with a normal diet were significantly lower than those kept on the BCR animal feed. By applying the SRCA method, we engendered the development of Ne/De mesoblastic renal tumours and My1/De leukaemia in all groups of experimental rats. No significant effects of the two tumour types were observed with regard to the lipid-raising impact of the BCR diet, nor did the diet affect the growing potential of the neoplasms. Therefore, we managed to establish a hypercholesterinaemic tumour-bearing investigational animal model that may usher in a new era in the assessment of the effects of a large number of chemopreventive molecules on only a small number of animals, which can be achieved fairly quickly and in a relatively affordable manner in preclinical circumstances.

## Figures and Tables

**Figure 1 ijms-23-13060-f001:**
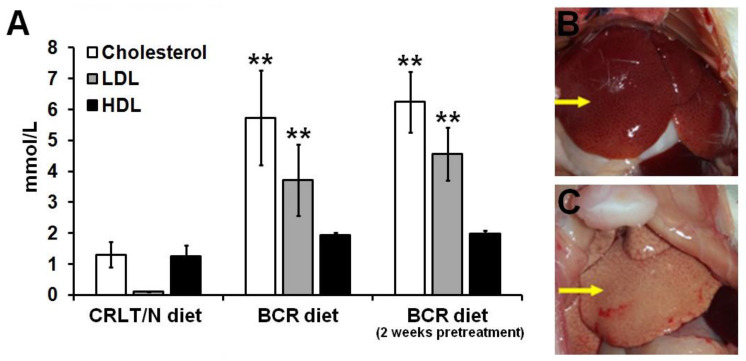
Effects of standard (CRLT/N) and BCR diets on healthy control rats. Quantitative analysis of the serum cholesterol fractions (**A**) after different feeding procedures. Autopsy images of the liver of healthy control rats fed with standard diet (**B**) and BCR diet (**C**). Arrow in (**B**) points to the liver of a healthy control rat fed with standard diet. Arrow in (**C**) shows the liver of a healthy control rat fed with BCR diet. Significance level (compared to the CRLT/N group): *p* ≤ 0.01 (**). CRLT/N: Charles River Laboratories feed/normal and BCR: butter and cholesterol rich.

**Figure 2 ijms-23-13060-f002:**
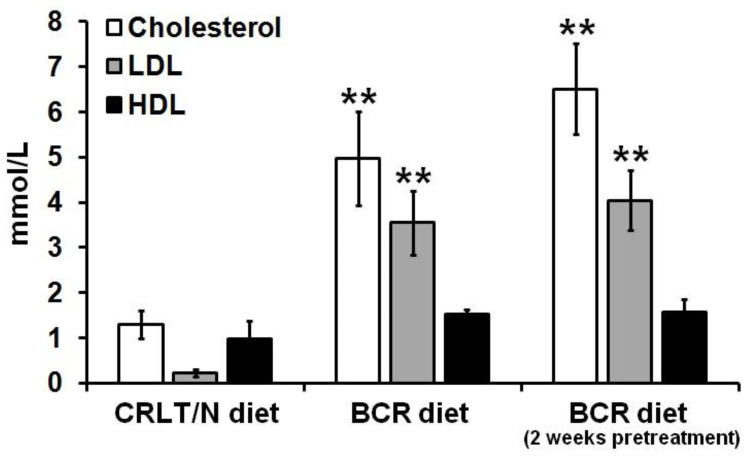
Effects of standard (CRLT/N) and BCR diets on the serum cholesterol fractions of Ne/De tumour-bearing rats. Significance level (compared to the CRLT/N group): *p* ≤ 0.01 (**). CRLT/N: Charles Rivers Laboratories feed/normal; BCR: butter and cholesterol rich; and Ne/De: mesoblastic nephroma.

**Figure 3 ijms-23-13060-f003:**
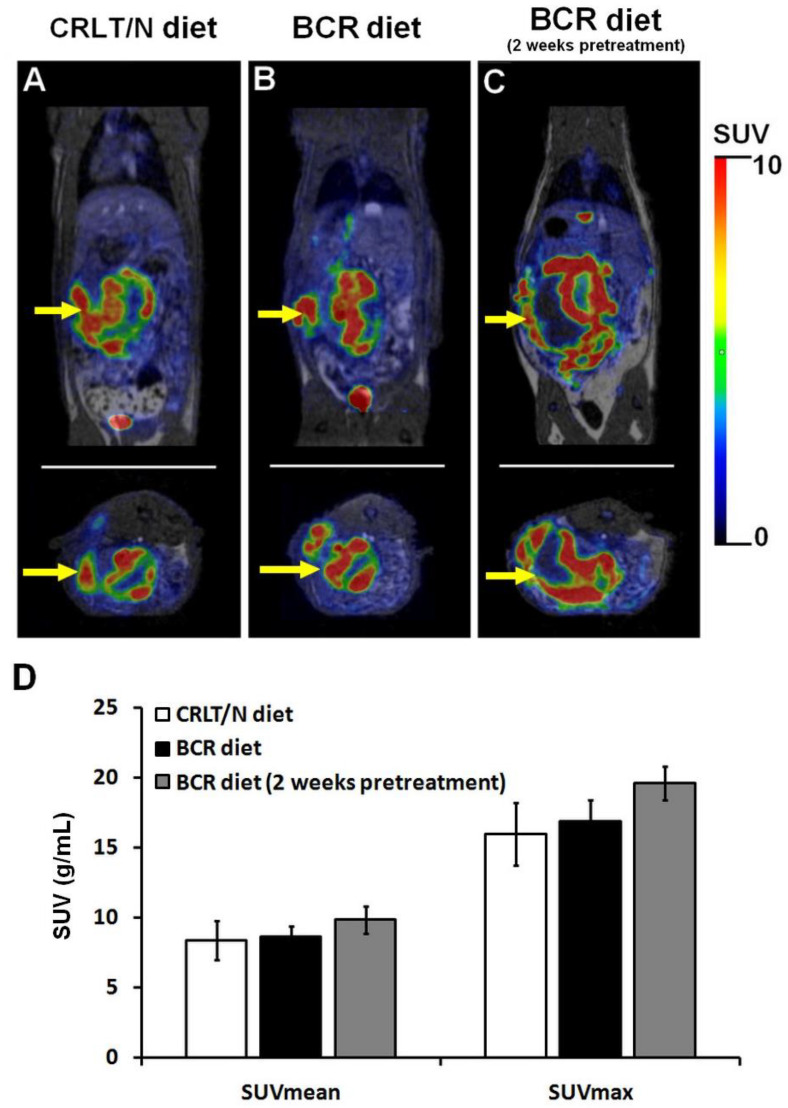
In vivo assessment of the impact of BCR diet on tumour growth using [^18^F]F-FDG PET/MRI examination. Coronal (upper row) and transaxial (lower row) decay-corrected PET/MRI images of the Ne/De tumour-bearing rats kept on (**A**) standard CRLT/N diet (group 4); (**B**) BCR diet (group 5); and (**C**) BCR diet with a two-week long pretreatment (group 6) are all demonstrated. Panel (**D**) represents the results of the quantitative SUV analysis of [^18^F]F-FDG uptake of the Ne/De tumours 50 min post-injection and 14 days after SRCA transplantation of Ne/De tumour cells. SUV values are presented as mean ± SD. Yellow arrow: Ne/De tumour. BCR: butter and cholesterol rich; [^18^F]F-FDG: 2-deoxy-2-[^18^F]fluoro-D-glucose; PET/MRI: positron emission tomography/magnetic resonance imaging; Ne/De: mesoblastic nephroma; CRLT/N: Charles Rivers Laboratories feed/normal; SUV: standardised uptake value; SRCA: subrenal capsule assay; and SD: standard deviation.

**Figure 4 ijms-23-13060-f004:**
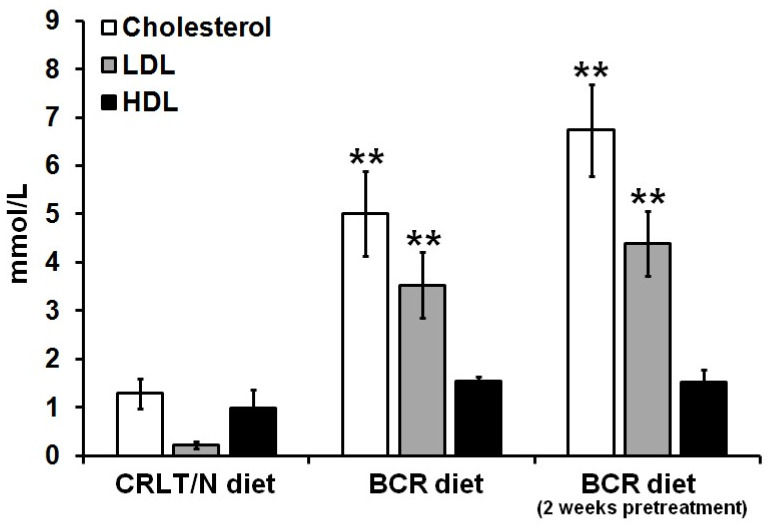
Effects of standard (CRLT/N) and BCR diets on the serum cholesterol fractions of My1/De tumour-bearing rats. Significance level (compared to the CRLT/N group): *p* ≤ 0.01 (**). CRLT/N: Charles Rivers Laboratories feed/normal; BCR: butter and cholesterol rich; and My1/De: myelomonoblastic leukaemia.

**Figure 5 ijms-23-13060-f005:**
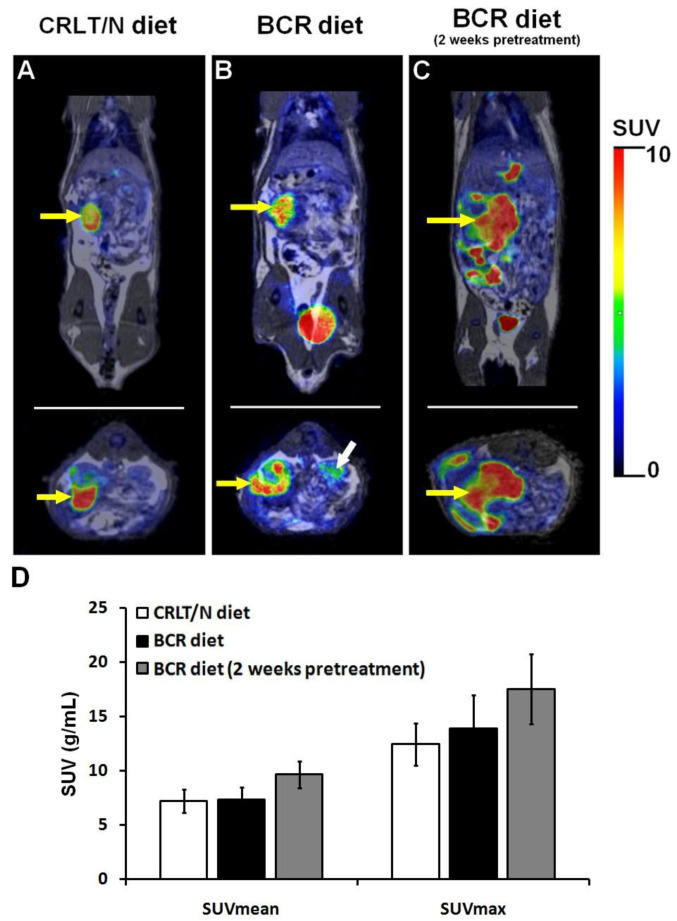
In vivo assessment of the impact of the BCR diet on tumour growth using [^18^F]F-FDG PET/MRI examinations. Coronal (upper row) and transaxial (lower row) decay-corrected PET/MRI images of My1/De tumour-bearing rats kept on (**A**) standard CRLT/N diet (group 4), (**B**) baseline BCR diet (group 5); and (**C**) BCR diet with a 2-week long pretreatment (group 6). Panel (**D**) represents the quantitative SUV analysis on [^18^F]F-FDG accumulation of My1/De tumours 50 min post-injection and 14 days after SRCA transplantation of the My1/De tumour cells. SUV values are presented as mean ± SD. Yellow arrows: My1/De tumour and white arrow: right kidney. BCR: butter and cholesterol rich; [^18^F]F-FDG: 2-deoxy-2-[^18^F]fluoro-D-glucose; PET/MRI: positron emission tomography/magnetic resonance imaging; My1/De: myelomonoblastic leukaemia; CRLT/N: Charles Rivers Laboratories feed/normal; SUV: standardised uptake value; SRCA: subrenal capsule assay; and SD: standard deviation.

**Figure 6 ijms-23-13060-f006:**
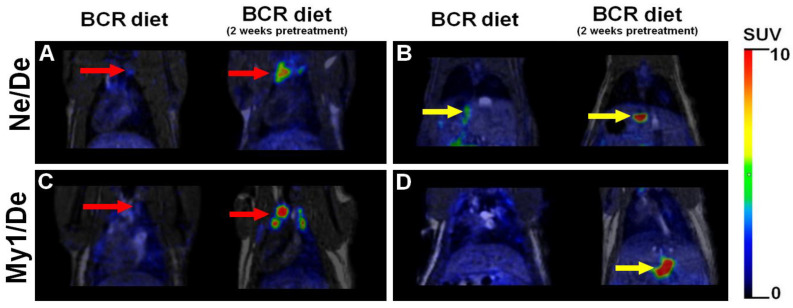
Representative decay-corrected coronal [^18^F]F-FDG PET/MRI images of metastases in tumour-bearing Ne/De and My1/De dyslipidaemic rats 50 min post-injection of [^18^F]F-FDG, and 14 days after SRCA-based tumour cell transplantation. Panels (**A**,**B**) demonstrate the PET/MRI images of parathymic lymph nodes and liver metastases, respectively in Ne/De tumour-bearing rats of group 5 (baseline BCR) and group 6 (pretreated BCR). PET/MRI images of parathymic lymph node (**C**) and liver metastases (**D**) of My1/De tumorous rats of group 5 (baseline BCR) and group 6 (pretreated BCR) are shown in panels C and D. Red arrows: metastatic parathymic lymph nodes and yellow arrows: liver metastases. [^18^F]F-FDG: 2-deoxy-2-[^18^F]fluoro-D-glucose; PET/MRI: positron emission tomography/magnetic resonance imaging; Ne/De: mesoblastic nephroma; My1/De: myelomonoblastic leukaemia; and SRCA: subrenal capsule assay.

**Table 1 ijms-23-13060-t001:** Comparison of the effects of standard (CRLT/N) and BCR diets on tumour growth in Ne/De tumour-bearing rats. Significance levels (compared to the CRLT/N group): *p* ≤ 0.05 (*) and *p* ≤ 0.01 (**).

Ne/De Tumorous Rats	Left Kidney (g)	Right Kidney (g)	Tumour Mass (g)
CRLT/N	7.46 ± 1.31	0.82 ± 0.20	6.74 ± 0.81
BCR diet	8.31 ± 0.73	0.75 ± 0.17	7.28 ± 0.73
BCR diet (2 weeks of pretreatment)	13.42 ± 1.09 *	0.78 ± 0.13	12.93 ± 1.27 **

CRLT/N: Charles Rivers Laboratories feed/normal; BCR: butter and cholesterol rich, Ne/De: mesoblastic nephroma.

**Table 2 ijms-23-13060-t002:** Comparison of the effects of standard (CRLT/N) and BCR diets on tumour growth in My1/De tumour-bearing rats. Significance levels (compared to the CRLT/N group): *p* ≤ 0.05 (*) and *p* ≤ 0.01 (**).

My1/De	Left Kidney (g)	Right Kidney (g)	Tumour Mass (g)
CRLT/N	3.22 ± 1.05	1.12 ± 0.24	2.47 ± 0.64
BCR diet	4.62 ± 0.89	0.99 ± 0.31	3.45 ± 0.39
pretreated BCR diet	7.46 ± 1.14 *	1.06 ± 0.24	6.26 ± 0.89 **

CRLT/N: Charles Rivers Laboratories feed/normal; BCR: butter and cholesterol rich; My1/De: myelomonoblastic leukaemia.

**Table 3 ijms-23-13060-t003:** Animals were subclassified into the following six experimental groups.

Groups	Status	Diet	Administration of Rodent Chow (Number of Weeks)
1	Healthy	Standard CRLT/N	2
2	Healthy	BCR	2 (baseline)
3	Healthy	BCR	2 + 2 (pretreatment + maintenance)
4	Tumorous	Standard CRLT/N	2
5	Tumorous	BCR	2 (baseline)
6	Tumorous	BCR	2 + 2 (pretreatment + maintenance)

**Table 4 ijms-23-13060-t004:** Components of the standard CRLT/N diet.

Standard CRLT/N Diet	
Component	Composition (%)
Dry matter	86.00
Crude protein	20.00
Digested protein	18.00
Lysine	0.97
Methionine	0.30
Methionine + cysteine	0.64
Crude fat	4.00
Crude fibre	4.30
Calcium	0.96
Phosphorus	0.67
Sodium	0.20

CRLT/N: Charles River Laboratories feed/normal.

**Table 5 ijms-23-13060-t005:** Components of the special butter and cholesterol-rich (BCR) diet.

BCR Diet	
Component	Composition (%)
Standard CRLT/N feed	60.00
Casein	9.50
Butter	20.00
Cholesterol	4.00
Sodium cholate	1.00
“SM8” salt mixture, according to Sós [29]	5.00
6-Methyl-2-thio-uracil	0.50

BCR: butter and cholesterol rich.

## Data Availability

The datasets used and/or analysed during the current study are available from the corresponding author upon reasonable request.

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
