# Peer review of "In Vivo Assessments of Mesoblastic Nephroma (Ne/De) and Myelomonoblastic Leukaemia (My1/De) Tumour Development in Hypercholesterolemia Rat Models"

_ijms, 2022, doi:10.3390/ijms232113060_

Round 1

Reviewer 1 Report

line 58: " a montain of evidence" this is exagerated. epidiomolgical data are more confusing and this is still a matter of debate and certainly not an evidence.

line 72: results from recent epidemiological studies on large population failed to show no protective effects of vitamines on all cancers.

I agree that your model is of interest for research

FDG allow the follow up of glycolytic tumors to be done. Does FDG labeling account for the whole tumor mass?

cholesterol by itself is not a tumor promoter. What about cholesterol metabolites? cholesteryl-fatty acid esters, colesterol 5,6-epoxides and thir metabolites oncosterone or even 27hydroxycholesterol? This is at least to be discussed.

The use of statin is very different in term of impact to your model in which we can suppose that the increase of circulating cholesterol level is due to the diet. But this remains to be demonstrated. If this is right than blocking cholesterol neosynthesis with statins. THis is very different  

Reviewer 2 Report

In general, the article fits the scope of the journal, it is well structured and the theme developed is interesting. The comments are favorable and intend to improve the article.

- The title is too long, try to keep it simple if possible!

- Why use the word hypercholesterinaemia instead of hypercholesterolemia throughout the text, which seems more objective to me. I suggest replacement.

- Confirm line 98 ((    )

- The idea of ​​creating animal models that can be used as models for therapy is undoubtedly an asset, although we know that the transition to the human does not always result in totality

- The objective seems to be very promising considering the results obtained

- The conclusions are in agreement with the evidence obtained and the possibility of testing chemotherapeutic molecules will be promising

Congratulations on the theme
